EMBO
Molecular Medicine

# Retinoic acid catabolizing enzyme CYP26C1 is a genetic modifier in SHOX deficiency

Antonino Montalbano[1], Lonny Juergensen[2], Ralph Roeth[1], Birgit Weiss[1], Maki Fukami[3], Susanne Fricke-Otto[4], Gerhard Binder[5], Tsutomu Ogata[6], Eva Decker[7], Gudrun Nuernberg[8,9], David Hassel[2] & Gudrun A Rappold[1,10,*]

## Abstract

Mutations in the homeobox gene *SHOX* cause SHOX deficiency, a condition with clinical manifestations ranging from short stature without dysmorphic signs to severe mesomelic skeletal dysplasia. In rare cases, individuals with SHOX deficiency are asymptomatic. To elucidate the factors that modify disease severity/penetrance, we studied a three-generation family with SHOX deficiency. The variant p.Phe508Cys of the retinoic acid catabolizing enzyme *CYP26C1* co-segregated with the *SHOX* variant p.Val161Ala in the affected individuals, while the *SHOX* mutant alone was present in asymptomatic individuals. Two further cases with SHOX deficiency and damaging *CYP26C1* variants were identified in a cohort of 68 individuals with LWD. The identified CYP26C1 variants affected its catabolic activity, leading to an increased level of retinoic acid. High levels of retinoic acid significantly decrease *SHOX* expression in human primary chondrocytes and zebrafish embryos. Individual morpholino knockdown of either gene shortens the pectoral fins, whereas depletion of both genes leads to a more severe phenotype. Together, our findings describe *CYP26C1* as the first genetic modifier for SHOX deficiency.

**Keywords** clinical variability; genetic modifiers; limb development; retinoic acid; skeletal dysplasia

**Subject Categories** Development & Differentiation; Genetics, Gene Therapy & Genetic Disease; Musculoskeletal System

## Introduction

In the last decade, many genetic variations responsible for the pathogenesis of disease have been identified. Next-generation sequencing and high-throughput genotyping platforms have revolutionized the pursuit of disease-causing genetic variants. However, disease phenotypes vary—even among individuals carrying the same causative genetic variants within one family—and our understanding of the factors that modify disease severity/penetrance remains incomplete.

Human height is a complex trait with a high heritability. Short stature affects approximately 3% of children worldwide and is diagnosed when height is significantly below the average of the general population for that person's age and sex. More precisely, short stature is statistically defined as two standard deviations (SD) below the mean population height for age, sex, and ethnic group (less than the third percentile) or, when evaluating shortness in relation to family background, more than two SD below the mid-parental height (Ranke, 1994). To date, many different etiologies of short stature are known and more than two hundred genes underlying growth control have been identified (Marchini *et al*, 2007; Durand & Rappold, 2013; Baron *et al*, 2015).

Mutations in the short stature homeobox-containing (*SHOX*) gene cause SHOX deficiency, the most frequent monogenic cause of short stature (Marchini *et al*, 2016). The gene is localized in the pseudoautosomal region 1 (PAR1) shared between the X and Y chromosomes. Its varied clinical manifestations include idiopathic/isolated short stature (ISS), Léri-Weill dyschondrosteosis (LWD), and Langer mesomelic dysplasia (LD). Heterozygous mutations in its coding or regulatory regions have been identified in up to 10% of patients diagnosed with ISS and 70% of patients with LWD, while homozygous mutations cause LD (Zinn *et al*, 2002; Rosilio *et al*, 2012; Marchini *et al*, 2016). SHOX deficiency also contributes to the

1 Department of Human Molecular Genetics, Heidelberg University, Heidelberg, Germany
2 Department of Internal Medicine III - Cardiology, Heidelberg University Hospital, Heidelberg, Germany
3 Department of Molecular Endocrinology, National Research Institute for Child Health and Development, Tokyo, Japan
4 Children's Hospital Krefeld, Krefeld, Germany
5 Children's Hospital, University of Tübingen, Tübingen, Germany
6 Department of Pediatrics, Hamamatsu University School of Medicine, Hamamatsu, Japan
7 Bioscientia Center for Human Genetics, Ingelheim, Germany
8 Center for Molecular Medicine, Cologne, Germany
9 Cologne Center for Genomics, Cologne, Germany
10 Interdisciplinary Centre for Neurosciences (IZN), University of Heidelberg, Heidelberg, Germany
*Corresponding author. Tel: +49 6221 565059; E-mail: gudrun.rappold@med.uni-heidelberg.de

short stature and skeletal features in Turner syndrome (Rao *et al*, 1997; Clement-Jones *et al*, 2000). Thus, SHOX deficiency is associated with a broad phenotypic spectrum ranging from short stature without dysmorphic signs to profound dysplasia (Rao *et al*, 1997; Belin *et al*, 1998; Shears *et al*, 1998). Patients with SHOX deficiency often present mesomelic (disproportionate) short stature, a selective shortening of the lower arms/legs. The phenotype may also include Madelung deformity (MD) of the wrist, considered to be the archetypal sign of LWD, which is characterized by shortening and bowing of the radius together with distal hypoplasia of the ulna. Skeletal manifestations are usually more severe in females than in males (Binder, 2011).

The clinical severity of SHOX deficiency varies even in family members carrying the same *SHOX* mutation (Schiller *et al*, 2000; Binder, 2011). In rare cases, family members with identical mutation present with stature within the normal range (Huber *et al*, 2006; Benito-Sanz *et al*, 2012; Bunyan *et al*, 2013). We postulated that SHOX deficiency provides a genetically sensitized background on which genetic modifiers may promote disease progression. To shed light on this phenomenon, we have analyzed a family with five affected individuals displaying LWD. All affected individuals carried a damaging missense mutation in the *SHOX* gene. Three family members with the same mutation were phenotypically unaffected. To explain this clinical variability, we hypothesized the presence of modifier gene(s). Using whole-genome linkage analysis and whole-exome sequencing, we identified a *CYP26C1* variant to segregate solely in the affected individuals and carried out subsequent genetic and functional assays to provide evidence for a modifying role of this gene.

# Results

### SHOX deficiency and clinical variability

We have analyzed a three-generation German family with five affected individuals displaying LWD (family 1). According to the pattern of transmission of the trait, a dominant inheritance was hypothesized (Fig 1A). All female patients (I:2, I:3, II:7, and III:2, Fig 1A) presented with mesomelic short stature with a SD between −3.14 and −4.51 as well as MD, the archetypal sign of LWD. The father (II:3) of the index patient (III:2) presented with mesomelic short stature with an SD of −2.63 and a borderline MD, consistent with the fact that males are less severely affected than females (Table EV1). The mother (II:4), an aunt (II:8), and the stepsister (III:1) of the index patient (III:2) also presented with short stature, but mesomelia and MD were not diagnosed (Table EV1). Sanger sequencing identified a heterozygous variant, c.482T>C (p.Val161Ala), in the *SHOX* gene (NG_009385) in all individuals affected with LWD. p.Val161 resides within the DNA binding domain (Fig 2A) and is highly conserved among *SHOX* vertebrate homologues. To determine the functional relevance of this variant, a luciferase assay in U2OS cells was carried out using the promoter of the *fibroblast growth factor receptor 3* (*FGFR3*) gene, which is a known SHOX target (Decker *et al*, 2011). We showed a strong influence of the variant on SHOX transcriptional activity (Fig 2A). However, three non-affected family members also carried the *SHOX* variant p.Val161Ala (III:4, III:5, and III:6, Fig 1A). Using multiplex ligation-

dependent probe amplification (MLPA), other major genetic lesions in PAR1, including the previously identified enhancer elements in the vicinity of the *SHOX* locus, were excluded in all individuals.

## Identification of *CYP26C1* as genetic modifier

To address putative genetic causes of this intra-familiar clinical variability, whole-genome linkage analysis was performed. This analysis defined a 2.02 Mb interval in the PAR1 of chromosome X (chrX:706800-2735491; hg19) with a total LOD score of 1.7 (Figs EV1 and EV2). Moreover, a LOD score of 2.4 was obtained in a 19.2 Mb region of chromosome 10 (chr10:85477515–104681710; hg19) (Fig EV3). This region encompasses 263 genes, but could not be further refined. Subsequently, whole-exome sequencing analysis was performed on the index patient (III:2) and her father (II:3). Variant filtering was performed according to the hypothesized dominant transmission of the phenotype (see Materials and Methods); 98 variants in 97 genes were obtained (Dataset EV1). All variants were tested using the *in silico* mutation prediction tools PolyPhen-2, Mutation Taster, SIFT, and PROVEAN, and 36 variants were predicted as disease causing/damaging by at least one of these programs.

Sanger sequencing of the 36 identified variants was performed in the five affected and in the eight non-affected family members where DNA was available. Only variants in two genes, *OPN4* (chr10:88419701-88419701; hg19) and *CYP26C1* (chr10:94828408-94828408; hg19), segregated with the phenotype. Both genes reside within the chromosome 10 interval, which was previously defined by linkage analysis. OPN4 (melanopsin) is a known photopigment of photosensitive retinal ganglion cells and was not considered further (Panda *et al*, 2002). *CYP26C1* (NG_007958.1) encodes for a member of the cytochrome P450 superfamily of enzymes involved in the catabolism of retinoic acid (RA) (Taimi *et al*, 2004; Uehara *et al*, 2006; Pennimpede *et al*, 2010). Loss-of-function analyses in mouse and zebrafish have shown that CYP26C1 RA catabolic activity is required for proper hindbrain development (Sirbu *et al*, 2005; Uehara *et al*, 2006; Hernandez *et al*, 2007). RA has been previously shown to play a key role in limb development (Cunningham & Duester, 2015) and *CYP26B1*, another member of the CYP26 family, is known to be involved in skeleton development (Laue *et al*, 2011). Moreover, RA has been previously shown to inhibit *Shox* expression in chicken limbs (Tiecke *et al*, 2006). Therefore, we decided to further investigate *CYP26C1* as a candidate modifier in *SHOX* deficient LWD patients.

### *SHOX* and *CYP26C1* are members of the RA pathway

The heterozygous missense variant c.1523T>G (p.Phe508Cys) in *CYP26C1* affected a highly conserved amino acid among vertebrates. p.Phe508Cys was predicted as damaging by all prediction tools that were applied (Dataset EV2). The variant has not yet been described in the three major public variants databases, Exome Aggregation Consortium (ExAC), Exome Variant Server (EVS), and 1000 Genomes Project (TGP). To assess whether this mutation affects the enzymatic activity of CYP26C1, we used the Cignal RARE system, a RA-responsive luciferase reporter assay, in U2OS cells treated with all-*trans* retinoic acid (ATRA) (Laue *et al*, 2011). Overexpression of wild-type CYP26C1 reduced the luciferase activity, confirming that it degrades ATRA. In contrast, the CYP26C1 p.Phe508Cys mutant was

**Figure 1.   Damaging variants in *SHOX* and *CYP26C1* in patients with LWD.**

A   Pedigree of family 1. Individuals II:3 and III:2 were analyzed by whole-exome sequencing. Damaging variants in *SHOX* and *CYP26C1* co-segregate with LWD.

B   Pedigrees of families 2 and 3.

Data information: filled symbol, LWD-affected individual; symbol with a slash, deceased individual; slash, divorced; red text, *SHOX* locus; green text, *CYP26C1* locus; +, wild-type allele; N.A., DNA not available; arrow, index patient.

not able to reduce luciferase activity, suggesting that this mutation impairs CYP26C1 enzymatic activity and consequently RA degradation (Fig 2B).

   Next, we tested *CYP26C1* expression in human primary chondrocytes, where *SHOX* is expressed. By performing RT–PCR and Western blot analysis, we could show that *CYP26C1* is expressed in these cells (Fig 3A). We then asked whether RA affected *SHOX* expression in human primary chondrocytes (Appendix Fig S1). Treatment of primary chondrocytes with 100 nM ATRA led to a significant reduction of *SHOX* levels (Fig 3B). RA exerts its function by regulating the transcriptional activity of nuclear retinoic acid receptors (RARs) that bind as heterodimers with retinoic X receptors (RXRs) to RA response elements. Binding of RA to these receptors triggers activation or repression of their target genes (Weston *et al*, 2003). To verify whether RA directly or indirectly regulates *SHOX* expression, we tested a previously described *SHOX* promoter luciferase reporter assay in U2OS cells (Verdin *et al*, 2015). Treatment

with ATRA led to a significant reduction of luciferase activity compared to mock control (Fig 3C). *In silico* analysis of the *SHOX* promoter predicted three putative RXRa binding sites (Fig 3C). However, disruption of these binding sites did not significantly influence the ATRA effect on the *SHOX* promoter, indicating an indirect effect of RA on *SHOX* expression (Fig 3C). Altogether, these results provide evidence that *CYP26C1* and *SHOX* are members of the same pathway: *CYP26C1* regulates *SHOX* expression by regulating intracellular levels of RA (Fig 3D).

### Additional genetic evidence in SHOX deficiency patients presenting with LWD

We then investigated whether the co-occurrence of damaging *SHOX* and *CYP26C1* variants in the severe phenotypes represents a unique finding specific to family 1. We screened *CYP26C1* in a cohort of 68 LWD individuals with proven SHOX deficiency and in a cohort of

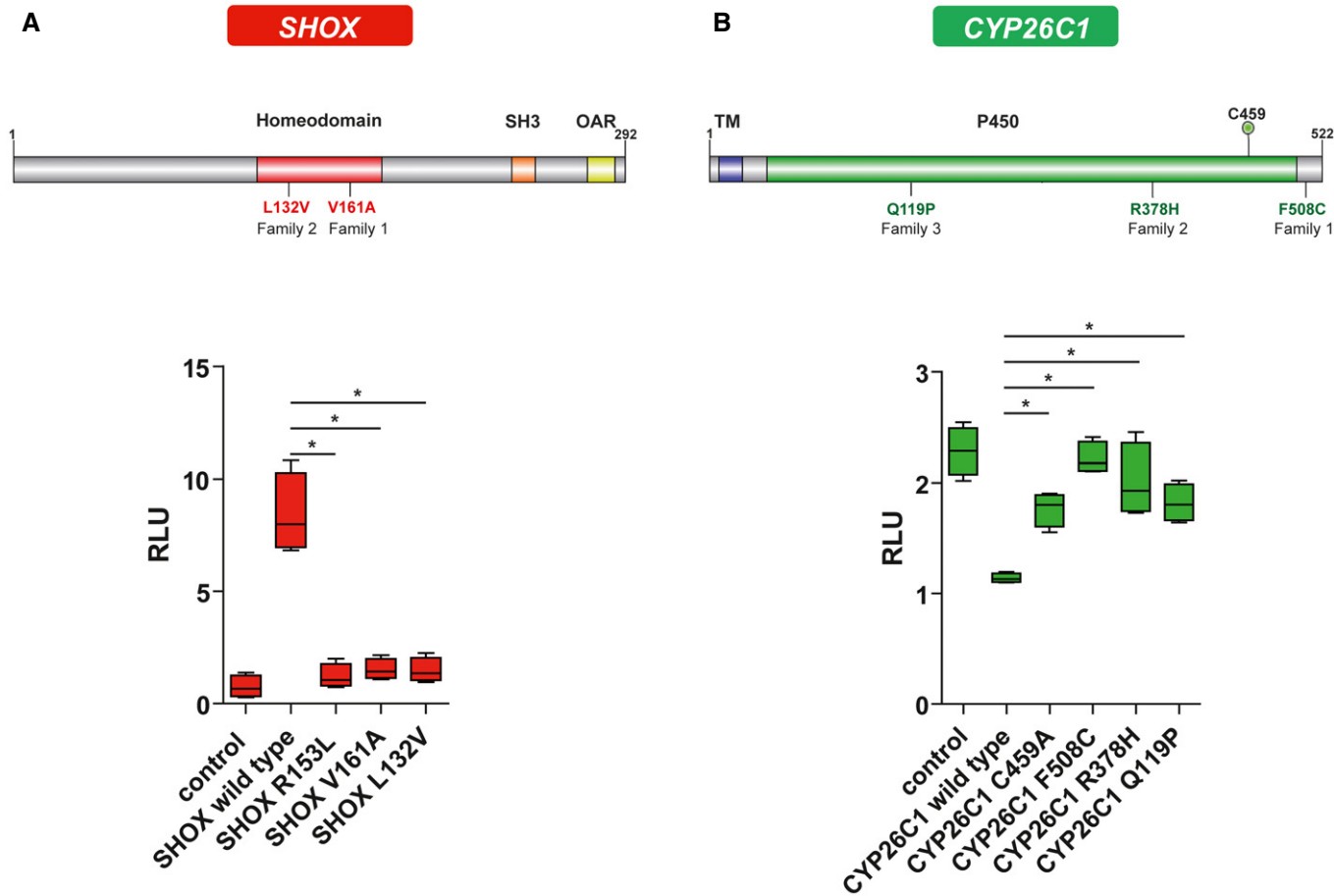

**Figure 2. Scheme of SHOX and CYP26C1 proteins and luciferase assays.**

A   SHOX protein representation and luciferase assay. The variants found in SHOX are indicated in red. Luciferase assays to test the impact of SHOX mutations on its transcriptional activity were performed on the *FGFR3* promoter in U2OS cells (*n* = 4). Experiments were performed in triplicates. pcDNA4-TO empty vector was used as control. R153L represents a mutation known to affect SHOX protein activity and was used as a positive control (Schneider *et al*, 2005). Homeodomain, DNA binding domain; SH3, Src homology 3 domain; OAR, OtpAristalessRax domain; RLU, relative light units.

B   CYP26C1 protein representation and luciferase assay. Variants found in CYP26C1 are indicated in green. Cignal RARE system luciferase assays to test the impact of CYP26C1 variants on its RA degradation activity were performed in U2OS cells treated with 250 nM all-*trans* retinoic acid (ATRA) for 24 h (*n* = 4). Experiments were performed in triplicates. pIRES2-EGFP empty vector was used as control. The residue C459 represents the iron binding residue (Q6V0L0, UniProtKB) and was mutated to Ala and used as a positive control. TM, transmembrane helix; P450, cytochrome p450 domain. RLU, relative light units.

Data information: The box represents the interquartile range. The whiskers represent Min to Max. *$P$-value = 0.0286, two-tailed Mann–Whitney non-parametric *t*-test.

350 control individuals with normal height. This analysis identified two further cases with co-occurrence of damaging variants in *SHOX* and *CYP26C1* in the patient cohort (Fig 1B and Dataset EV2). No functional *CYP26C1* damaging variants were found in the control individuals with normal height (Dataset EV3; Appendix Fig S2).

In family 2, the affected daughter (II:2) carried a heterozygous missense variant in *SHOX*, c.349C>G (p.Leu132Val). This variant has been shown to affect SHOX homodimerization and DNA binding (Schneider *et al*, 2005). The father (I:1) and the sister (II:1) also carried this *SHOX* variant, but they were reported to be unaffected without dysmorphic signs. Screening of *CYP26C1* identified a heterozygous missense variant c.1133G>A (p.Arg378His) only in the daughter with LWD (II:2) (Fig 1B, Table EV1). DNA from the mother (I:2) was not available for the *CYP26C1* gene analysis. The third case was an affected girl who carried a *de novo* heterozygous deletion of *SHOX* and a missense variant in *CYP26C1*, c.356A>C

(p.Gln119Pro) (Table EV1). Both parents and the brother presented with normal stature and had no dysmorphic signs. The *CYP26C1* variant was inherited from the father. Since the *SHOX* deletion is *de novo*, this family cannot demonstrate co-segregation, although it adds to the overall evidence that damaging variants in *SHOX* and *CYP26C1* co-occur in individuals with severe LWD phenotypes.

We then performed luciferase analysis of all the *SHOX* and *CYP26C1* mutations and could demonstrate their negative impact on protein activity (Fig 2B). We conclude that in addition to family 1, two out of 68 LWD patients with SHOX deficiency presented with functional damaging mutations in *CYP26C1*, while no damaging mutations with impact on protein activity were identified in 350 control individuals with normal height. In all families where clinical information was available, all individuals with damaging mutations in both *SHOX* and *CYP26C1* presented with short stature and severe skeletal phenotypes.

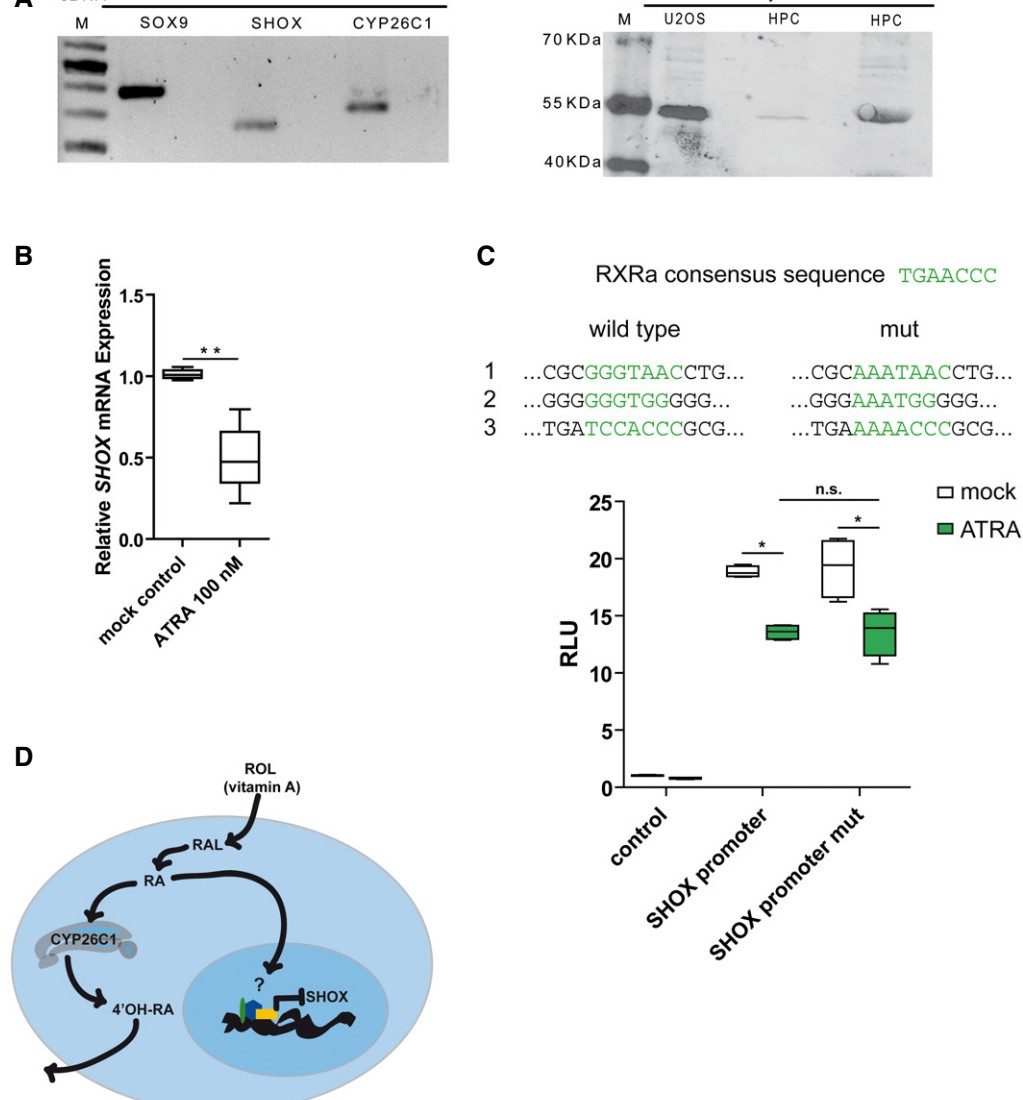

**Figure 3.  CYP26C1 is expressed in human primary chondrocytes and retinoic acid affects SHOX expression.**

A   Left panel, RT–PCR showing the expression of *CYP26C1* mRNA in human primary chondrocytes. *SOX9* was used as chondrocyte marker. Right panel, Western blot showing the expression of CYP26C1 in human primary chondrocytes on protein level. Overexpression of CYP26C1 in U2OS cells was carried out to compare protein sizes. +, cDNA; −, water; M, marker; HPC, human primary chondrocytes.

B   Relative expression of *SHOX* mRNA normalized to the reference genes *SDHA* and *HPRT* in human primary chondrocytes treated with ATRA 100 nM for 6 h (*n* = 5). One outlier with high relative expression (2.3-fold) was excluded from ATRA 100 nM treatment (two-sided Grubbs' test, *Z*-value 2.34, *P*-value < 0.05).

C   *SHOX* promoter was cloned in pGL3basic for luciferase reporter experiments as previously described (Verdin *et al*, 2015). *In silico* analysis of *SHOX* promoter identified three putative RXRa binding sites which were mutated to test their direct effect on *SHOX* expression upon treatment with ATRA 250 nM in U2OS cells (*n* = 4). Experiments were performed in triplicates. RLU, relative light units.

D   *CYP26C1* and *SHOX* are members of the retinoic acid pathway. Vitamin A, retinol (ROL), enters the cell and is oxidized to retinaldehyde (RAL). RAL is then oxidized to retinoic acid (RA). RA can enter the nucleus and regulate the expression of its targets. CYP26C1 controls RA intracellular levels by oxidizing this molecule in more hydrosoluble retinoid molecules like 4′-hydroxy-retinoic acid (4′-OH-RA), which can be readily excreted. High levels of RA downregulate *SHOX* expression.

Data information: The box represents the interquartile range. The whiskers represent Min to Max. (B)  **P = 0.0079, (C) *P-value = 0.0286, n.s., not significant, (B, C) two-tailed Mann–Whitney non-parametric *t*-test.

### Modeling SHOX deficiency in zebrafish embryos

Finally, to gain insight into the role of *SHOX* and *CYP26C1* interaction on limb development, we performed antisense morpholino (MO) knockdown experiments in zebrafish embryos. Testing of

each MO proofed efficacy (Appendix Figs S4–S6). We first assessed the effect of *shox* reduction on limb/fin development. Upon *shox* knockdown, the zebrafish embryos showed an overall delayed growth and a strong impairment of the developing pectoral fins, in concordance with a previous report (Sawada *et al*, 2015; Fig 4).

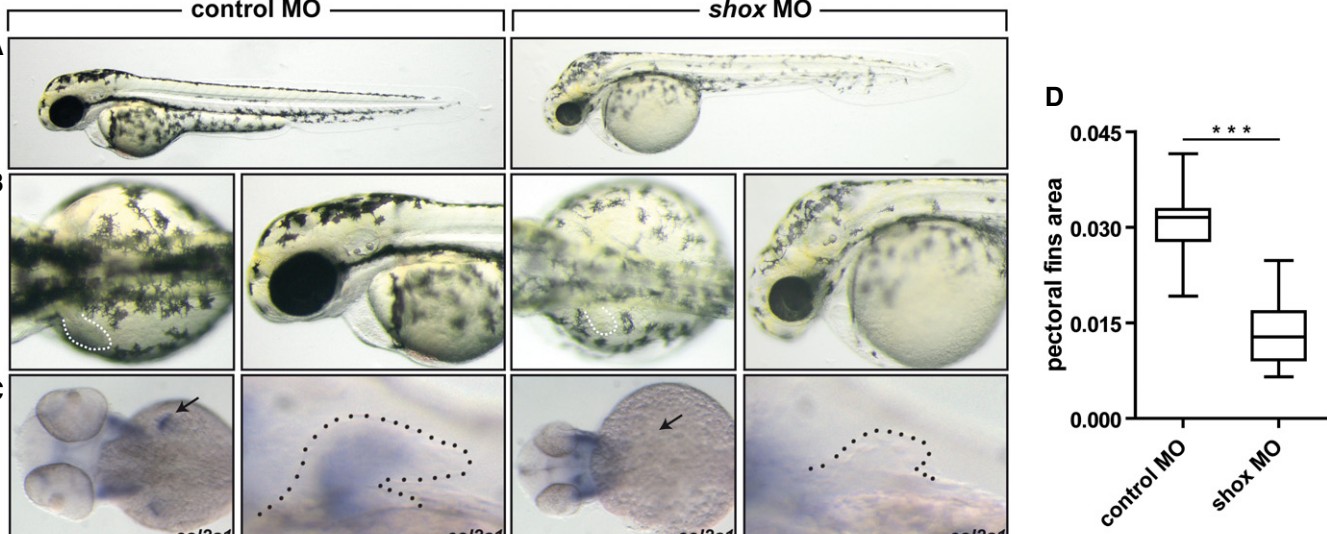

**Figure 4.  Pattern of defects in zebrafish embryos injected with anti-*shox* morpholino.**

Wild-type embryos injected with control MO or with *shox* MO.

A   Lateral views of the embryos at 55 hours post-fertilization (hpf).

B   Dorsal view and magnification on the lateral view of the embryos. Dotted line, pectoral fins. *shox* morphants show smaller fins compared to controls (*n* = 30 embryos).

C   Expression of *col2a1* at 55 hpf was examined by *in situ hybridization* in wild-type embryos injected with control MO and with *shox* MO. Arrow and dotted line indicate the pectoral fin.

D   Pectoral fin area was measured by ImageJ (*n* = 30 embryos).

Data information: The box represents the interquartile range. The whiskers represent Min to Max. ***P-value = 0.0001, two-tailed unpaired Student's *t*-test.

Using whole-mount *in situ* hybridization, we showed that *col2a1* expression, an established marker of chondrocytes, was dramatically reduced in the pectoral fin buds upon *shox* reduction (Fig 4C). Overexpression experiments in zebrafish embryos of human *SHOX* wild type and the variants identified in families 1 and 2 corroborated the functional significance of SHOX p.Val161Ala (family 1) and p.Leu132Val (family 2) (Appendix Fig S7).

We then analyzed the effect of *cyp26c1* knockdown in zebrafish embryos. Expression of *cyp26c1* in the pectoral fins has been previously demonstrated by whole-mount *in situ* hybridization (Gu *et al*, 2005). Knockdown of *cyp26c1* resulted in a significant reduction of pectoral fin size, although less strikingly compared to the *shox* knockdown (Fig 5B–E). The embryos also showed an abnormal development of the otic vesicles and pharyngeal arches (Fig 5B). As reported previously, we also found signs of cardiac dysfunction as evident by the pericardial edema (Rydeen & Waxman, 2014) (Fig 5B). In the pectoral fins, we demonstrated that the expression of *col2a1* was strongly reduced or absent (Fig 5C). In addition, the expression of *shox* was shown to be decreased upon *cyp26c1* knockdown (Fig 5D and G). MO knockdown of *cyp26c1* led to a significant increase in RA levels compared to control MO, which is in line with its role in RA degradation (Fig 5F). We treated zebrafish embryos with RA and obtained a significant downregulation of *shox* expression, further corroborating the hypothesis of *SHOX* as a component of the RA pathway (Fig 5H). Finally, we tested the functional significance of the *CYP26C1* variants identified in families 1–3 in zebrafish embryos and could show a striking effect on CYP26C1 activity (Appendix Fig S8). Together, these data indicate that *CYP26C1* is involved in limb development and acts by controlling RA levels.

To test the hypothesis that *CYP26C1* is a modifier of SHOX deficiency, we tested different concentration of *shox* and *cyp26c1* MOs to determine subphenotypic dosages. While single knockdown of either *shox* or *cyp26c1* using subphenotypic MO doses did not result in any obvious phenotype, double knockdown (simultaneous injection) of *shox* and *cyp26c1* produced significantly smaller pectoral fins (Fig 6A–D). The relative expression of *shox* mRNA expression was also significantly reduced (Fig 6E). Staining for *col2a1* revealed impaired pectoral fin development (Fig 6B). These data further corroborate the hypothesis that *CYP26C1* damaging mutations contribute to severe phenotypes in *SHOX* deficient individuals.

## Discussion

Variable phenotypes can be attributed to genetic and environmental factors. Our study describes a Mendelian disorder with wide phenotypic variability, which provides a unique opportunity to identify the genetic causes of variability. Individuals with SHOX deficiency but normal stature in a three-generation family (Family 1) prompted us to postulate genetic modifiers as the possible reasons of this variability. Three individuals with height within the normal range and neither mesomelia nor MD had inherited the same *SHOX* variant p.Val161Ala as their five affected relatives with LWD. By combining whole-genome linkage and whole-exome sequencing analysis in family 1, we identified *CYP26C1* as the gene co-segregating with the clinical phenotype and hypothesized a putative modifier function on *SHOX* expression. The mother (II:4), an aunt (II:8), and the stepsister (III:1) of the index patient (III:2) presented with short stature,

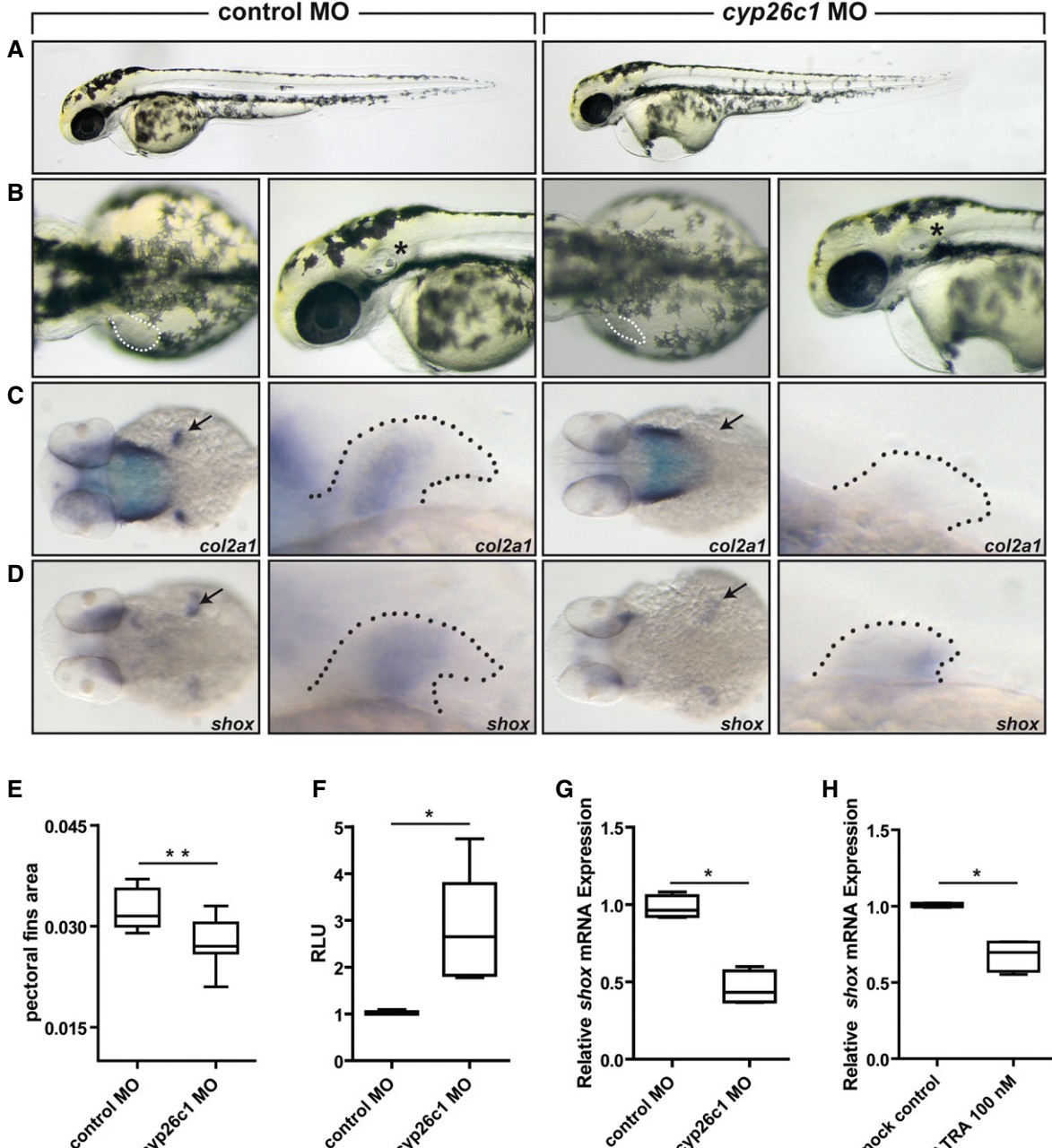

**Figure 5.  Pattern of defects in zebrafish embryos injected with anti-*cyp26c1* morpholino.**
Wild-type embryos injected with control MO or with *cyp26c1* MO.

A, B     (A) Lateral views of the embryos at 55 hours post-fertilization (hpf). (B) Dorsal view and magnification on the lateral view of the embryos. Dotted line, pectoral fins; *, otic vesicle. *cyp26c1* morphants show smaller fins compared to controls (*n* = 30 embryos).

C, D     Expression of *col2a1* (C) and *shox* (D) at 55 hpf was examined by *in situ* hybridization in embryos injected with control MO or with *cyp26c1* MO. (C) Dorsal view and magnification on the pectoral fins of *col2a1* expression. Arrow and dotted line indicate the pectoral fin. (D) Dorsal view and magnification on the pectoral fins of *shox* expression. Arrow and dotted line indicate the pectoral fin.

E       Pectoral fin area was measured by ImageJ (*n* = 30 embryos).

F       Cignal-RARE system luciferase assay to test *cyp26c1* MO knockdown effect on RA acid levels in zebrafish embryos (*n* = 5 replicates). In each replicate, 20–30 embryos per condition were assayed. RLU, relative light units.

G       Relative *shox* mRNA expression normalized to reference genes *eef1a* and *b-actin* in zebrafish embryos injected with control MO or *cyp26c1* MO (*n* = 4). RNA was extracted from 10–15 injected embryos at 55 hpf.

H       Relative *shox* mRNA expression normalized to reference genes *eef1a* and *b-actin* in zebrafish embryos treated with 100 nM ATRA (*n* = 4). Embryos were collected at 24 hpf and treated with mock control or ATRA. RNA was extracted from 10–15 embryos after 6-h treatment.

Data information: The box represents the interquartile range. The whiskers represent Min to Max. (E) **P-value = 0.0006, (F) *P-value = 0.0119, (G) *P-value = 0.0286, (H) *P-value = 0.0286. (E) Two-tailed unpaired Student's *t*-test. (F–H) Two-tailed Mann–Whitney non-parametric *t*-test.

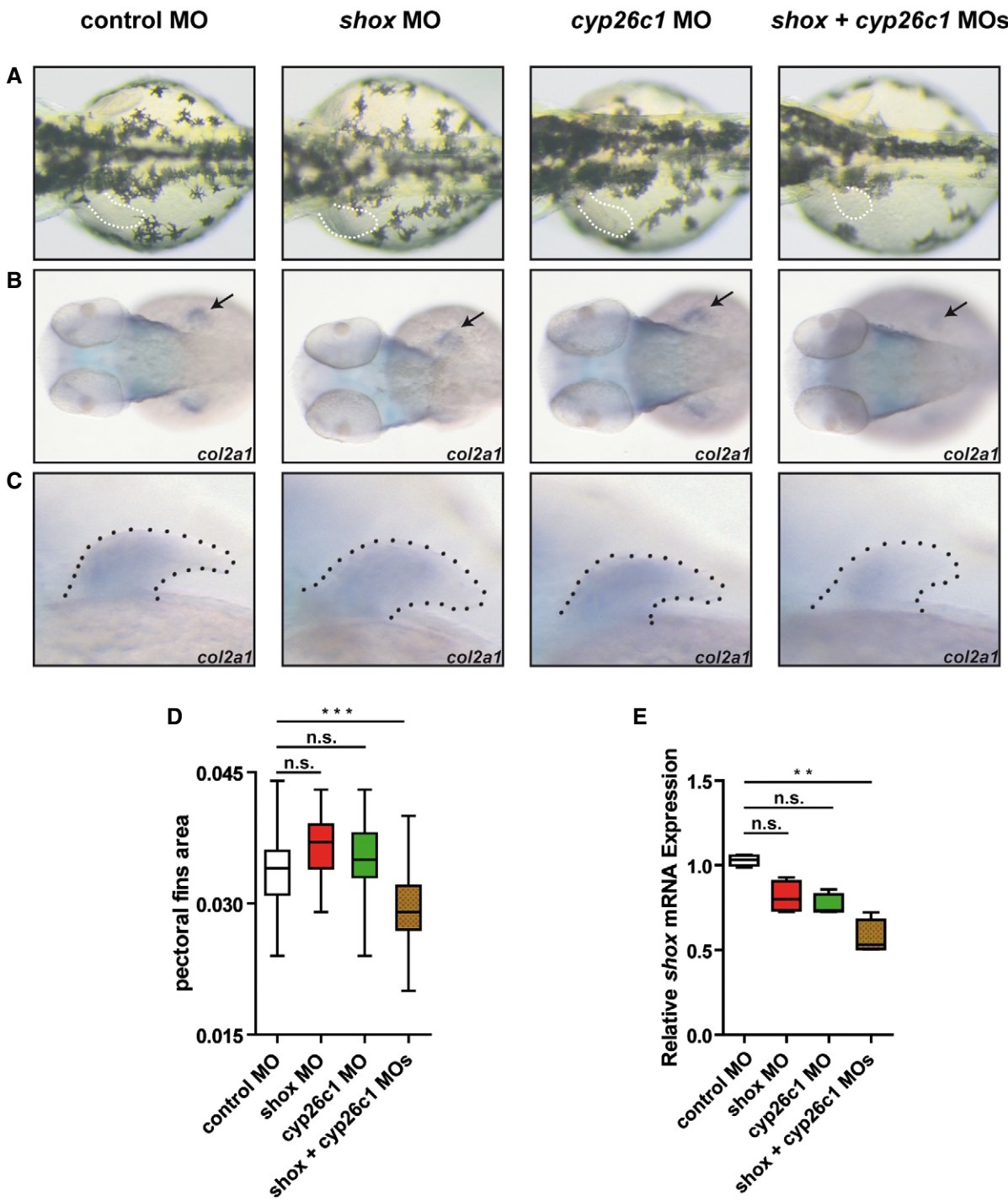

**Figure 6.  Co-injection of titrated subphenotypic doses of anti-*shox* and anti-*cyp26c1* morpholinos impairs limb development.**

Wild-type embryos injected with control MO, subphenotypic doses of *shox* MO, *cyp26c1* MO, or a combination of *shox* + *cyp26c1* MOs.

A    Dorsal views of the embryos at 55 hours post-fertilization (hpf). Dotted line, pectoral fins. *shox* + *cyp26c1* double morphants show smaller fins compared to control and single MOs (*n* = 30 embryos).

B, C    Dorsal view and magnification on the pectoral fins of *col2a1* expression at 55 hpf. Arrow and dotted line indicate the pectoral fin.

D    Pectoral fin area was measured by ImageJ (*n* = 30 embryos). Data are shown as means ± SD. ***$P$ = 0.0001 (one-way ANOVA with Bonferroni's multiple comparison test).

E    Relative expression of *shox* mRNA normalized to reference genes *eef1a* and *b-actin* in zebrafish embryos injected with control MO, *shox* MO, *cyp26c1* MO, or *shox* + *cyp26c1* MOs (*n* = 4). RNA was extracted from 10–15 injected embryos at 55 hpf. Data are shown as means ± SD. **$P$ = 0.0048 (Kruskal–Wallis with Dunn's multiple comparison test).

Data information: The box represents the interquartile range. The whiskers represent Min to Max. (D) ***$P$-value = 0.0001, (E) **$P$-value = 0.0048. (D) One-way ANOVA Bonferroni's multiple comparison test. (E) Klustal–Wallis Dunn's multiple comparison test.

but dysmorphic skeletal signs were not diagnosed. These individuals did not carry damaging variants neither in *SHOX* nor in *CYP26C1*, suggesting that different factors contribute to their short stature phenotype. Mesomelia and MD were present only in those individuals carrying damaging variants in both *SHOX* and *CYP26C1*.

Defective *CYP26C1* and *SHOX* were found to lead to a more severe phenotype in two further unrelated patients with LWD (Fig 1B). No functionally damaging variants in *CYP26C1* were found in 350 control individuals with normal height, suggesting that this co-occurrence is not coincidental (*P*-value = 0.0261, two-tailed Fisher's exact test). In addition, we browsed TGP for genotypes bearing both *SHOX* and *CYP26C1* damaging variants (predicted as damaging by at least one of the prediction tools used). A limitation of this approach is that not for all variants individual genotypes are available. We could not find any individual from the TGP database carrying damaging variants in both genes (1000 Genomes Project Consortium, 2015). Finally, we browsed the ExAC database and estimated the frequency of *SHOX* and *CYP26C1* damaging variants (Lek *et al*, 2016). To estimate the frequencies, we summed the allele frequencies of each damaging variant (predicted as damaging by at least one of the prediction tools used) reported in the ExAC database. The estimated allele frequencies were 0.3 and 0.6% for *SHOX* and *CYP26C1*, respectively. With these estimated frequencies, the probability for an individual to bear a damaging variant in both genes is $1.8 \times 10^{-5}$.

CYP26C1 is an enzyme belonging to the cytochrome P450 superfamily and is involved in the oxidation of RA to generate polar retinoid species, which are readily excreted. Thus, CYP26C1 is involved in the catabolism of RA (Taimi *et al*, 2004; Uehara *et al*, 2006; Pennimpede *et al*, 2010). Accordingly, we found that damaging mutations in CYP26C1 reduce its RA catabolizing activity leading to a higher concentration of this retinoid in U2OS cells (Fig 2B).

Retinoic acid plays a key role in development, including formation of the body axis and skeleton (Pennimpede *et al*, 2010; Cunningham & Duester, 2015). During skeletal development, RA coordinates the development of central body axis, limb axis, and cranium. Moreover, it controls chondroblast differentiation and coordinates maturation and replacement of bone tissue during endochondral ossification (Weston *et al*, 2003). RA also plays a role in the postnatal maintenance of bone (Green *et al*, 2015). An excess or deficiency of RA dysregulates the expression of the respective target genes, which has a dramatic effect on development (Weston *et al*, 2003). Excess or loss of RA has, for example, been shown to impair pectoral fin development in zebrafish and results in reduced forelimb size or complete loss of forelimb in mouse embryos (Uehara *et al*, 2006; Akimenko & Ekker, 1995; Begemann *et al*, 2001; Emoto *et al*, 2005; Cunningham *et al*, 2013). Hence, a tight regulation of RA metabolism is essential.

We demonstrated for the first time that *CYP26C1* is expressed on RNA and protein level in human primary chondrocytes (Fig 3), suggesting a role for *CYP26C1* in the fine regulation of RA in these cells. Damaging mutations in CYP26C1 that affect its RA oxidation activity may therefore lead to high levels of this retinoid. In support of this hypothesis, we found that knockdown of *cyp26c1* in zebrafish embryos increased RA levels and reduced *col2a1* expression (Fig 5F). Previous experiments on chicken limbs have shown that treatment with excess RA strongly reduces *Shox* expression (Tiecke

*et al*, 2006). Consistent with a role of RA in *SHOX* regulation, we have shown that treatment of human primary chondrocytes and zebrafish embryos with 100 nM RA significantly reduced *SHOX* expression, while this was not the case at 10–50 nM RA (Fig 2B and 5H; Appendix Fig S1). Luciferase reporter analyses of the human *SHOX* promoter suggest an indirect effect of RA on *SHOX* expression (Fig 3C). RA could lead to *SHOX* downregulation independently of RAREs, for example, by controlling the activation of other transcription factors, by non-classical associations of receptors with other proteins, or other mechanisms (Balmer & Blomhoff, 2002). Finally, we showed that reduction of *cyp26c1* in zebrafish embryos leads to the downregulation of *shox* expression (Fig 5D and G). Taken together, our results suggest that *SHOX* expression is not normally affected by endogenous RA in the nanomolar range (De Leenheer *et al*, 1995; Cunningham *et al*, 2013); however, loss of *CYP26C1* results in excess RA that downregulates *SHOX* expression.

Height depends mostly on the longitudinal growth of the limbs. Limbs contain long bones that are formed through endochondral ossification. This process involves the aggregation of mesenchymal progenitors that differentiate into chondrocytes, which form a cartilage template that is then replaced by bone tissue. Endochondral ossification is orchestrated by a complex network of signaling pathways and transcription factors and is highly conserved in vertebrates. *SHOX* is one of the genes involved in this network (Blaschke & Rappold, 2001). In accordance with a role for *SHOX* in limb development, we could demonstrate that *shox* reduction impairs pectoral fin development in zebrafish embryos (Fig 4).

Gene dosage of the *SHOX* locus has previously been shown to determine height: SHOX deficiency causes short stature, whereas *SHOX* overdosage has been linked to tall stature in Klinefelter syndrome (Ogata *et al*, 2001). SHOX has been suggested to repress growth plate fusion and skeletal maturation in the distal limbs. Unlike SHOX deficiency, which is characterized by premature epiphyseal plate fusion and relatively advanced skeletal maturation, *SHOX* overdosage leads to a delayed growth plate fusion and consequently to longer limbs (Ogata *et al*, 2001). Thus, different dosages of the SHOX protein play a role in determining the adult height of an individual.

We have found that damaging mutations in *CYP26C1* increase RA levels, which affect *SHOX* dosage and exacerbate SHOX deficiency phenotypes. To corroborate this hypothesis, we have demonstrated that the co-injection of subphenotypic dosages of *shox* and *cyp26c1* MOs strongly impaired pectoral fins (Fig 6). However, it is possible that damaging mutations in *CYP26C1* modify SHOX deficiency not only by directly affecting its expression, but also by misregulating other genes involved in the formation of long bones.

Homozygous/compound heterozygous mutations in *CYP26C1* have previously been associated with focal facial dermal dysplasia type IV (Slavotinek *et al*, 2013), a mild disorder of the skin. However, different *CYP26C1* variants were reported and information on height in these individuals was not available. MO knockdown experiments of *cyp26c1* in zebrafish embryos revealed abnormal pectoral fin development, but also an impairment of the otic vesicle and pharyngeal arches, structures that encompass the sites of the skin lesions observed in focal facial dermal dysplasia type IV (Slavotinek *et al*, 2013) (Fig 5B).

In 1963, Grüneberg (1963) defined modifiers as "genes capable of modifying the manifestation of a mutant gene without having an obvious effect on the normal condition". It would therefore be interesting to find out whether damaging *CYP26C1* variants have detectable phenotypes in the absence of SHOX deficiency. In family 3, the father (II:1), for example, carries a *CYP26C1* damaging variant but presented height within the normal range and no obvious dysmorphic signs. A systematic analysis on a large cohort of patients with or without dysmorphic signs and short stature is therefore needed to resolve whether *CYP26C1* variants alone can contribute to short stature phenotypes.

Taken together, this study represents an effort to elucidate the genetic causes of variability on the clinical manifestation in patients with SHOX deficiency. The RA pathway and one of its components, *CYP26C1*, were uncovered as biological triggers that we suggest to modify the severity of SHOX deficiency and alter the course of this disease. Only a few genetic modifiers have been reported in the literature so far (Oprea *et al*, 2008; Ebermann *et al*, 2010; Bečanović *et al*, 2015; Corvol *et al*, 2015; Guo *et al*, 2015; Lee *et al*, 2015), but recent technological advances are helping to broaden the understanding of the molecular basis of quantitative or discrete qualitative differences in phenotype. A primary reason to identify disease modifiers is to enable the accurate prediction of disease progression and improve therapeutic development. In the case of SHOX deficiency, manipulating the RA signaling pathway may be therapeutically beneficial.

# Materials and Methods

## Patients and controls

### Family 1

Family 1 comprised 17 German individuals with five affected members diagnosed with LWD. DNA was available from 14 individuals (I:1, I:2, I:3, II:3, II:4, II:6, II:7, II:8, III:1, III:2, III:3, III:4, III:5, III:6). Four females (I:2, I:3, II:7, III:2) presented with mesomelia and MD. In the affected male (II:3), mesomelia and MD were borderline. Individuals (II:4, II:8, III:1) presented with short stature but no dysmorphic signs. The index patient (III:2) was diagnosed with LWD at the age of 8 years and treated with growth hormone from the age of 8.9 years onwards. Age of menarche was of 12 years. Further details on the clinical data are given in Table EV1.

### Cohort of patients with SHOX deficiency

The sample comprised 68 unrelated cases with LWD and proven SHOX deficiency; it comprised 60 Europeans (28 Germans) and 9 Japanese. Two out of the 68 cases carried damaging mutations in *CYP26C1* and all the available family information was retrieved (see below).

Family 2 comprised four French people with one daughter affected with LWD; parents and sister were unaffected without dysmorphic signs. The affected daughter (II:2) presented with short stature, mesomelia, and MD (available clinical data are given in Table EV1).

Family 3 comprised four Japanese individuals with one daughter presenting with LWD. Parents and brother were reported with normal stature and no dysmorphic signs (available clinical data are given in Table EV1).

### Controls

As controls, we screened 350 (240 Germans, 110 Japanese) individuals with normal stature and no dysmorphic signs. We also compared frequencies to publicly available databases (ExAC, TGP, and EVS).

## Linkage analysis

We genotyped DNA samples from 12 family members of family 1 using the Affymetrix GeneChip Human Mapping 50K Xba 240 Array (Affymetrix). Genotypes were called by the GeneChip® DNA Analysis Software (GDAS v3.0, Affymetrix). We verified sample genders by counting heterozygous SNPs on the X chromosome. Relationship errors were evaluated with the help of the program Graphical Relationship Representation (Abecasis *et al*, 2001). The program PedCheck was applied to detect Mendelian errors (O'Connell & Weeks, 1998) and data for SNPs with such errors were removed from the dataset. Non-Mendelian errors were identified by using the program MERLIN (Abecasis *et al*, 2002) and unlikely genotypes for related samples were deleted. Non-parametric linkage analysis using all genotypes of a chromosome simultaneously was carried out with MERLIN.

Parametric linkage analysis was performed by the program ALLEGRO (Gudbjartsson *et al*, 2000), assuming dominant inheritance with complete penetrance and a disease allele frequency of 0.0001. Haplotypes were reconstructed with ALLEGRO and presented graphically with HaploPainter (Thiele & Nürnberg, 2005). All data handling was performed using the graphical user interface ALOHOMORA (Rüschendorf & Nürnberg, 2005). Using ALLEGRO, we identified one genomic region of 19.2 Mb on chromosome 10 (chr10:85477508-101462931; hg19) with LOD score of 2.4 (Figs EV1 and EV3). The LOD score obtained was the maximum LOD score expected in this family. A second peak with LOD score of 1.7 localized in a 2.02 Mb region of PAR1 (chrX:706800-2735491; hg19) (Fig EV2). The reason for the reduced LOD score in PAR1 is individuals III:4 and III:6 who carry the disease haplotype although they are unaffected (Fig EV2).

## Whole-exome sequencing

Whole-exome sequencing was performed as previously described (Haack *et al*, 2012). Exomes were enriched in solution provided by the manufacturer and indexed with SureSelect, XT Human All Exon 50 Mb kits (Agilent). Samples were sequenced as 100-bp paired-end runs on a HiSeq2000 system (Illumina). Read alignment to the human genome assembly was performed with the Burrows-Wheeler Aligner (v.0.5.8.). SAMtools (v.0.1.7.) was used for detecting single nucleotide variants and small insertions and deletions. We generated 8.8 and 11.2 gigabases of sequence resulting in 89 and 93% of the target regions being covered at least 20 times for index patient (III:2) and her father (II:3), respectively. To identify putative candidate genes based on dominant inheritance, we filtered all variants that were present in both individuals, but absent in all except two of 1,297 in-house control exomes.

## Sanger sequencing

Sanger sequencing was performed on MegaBACE sequencer using the DYEnamic™ ET Terminator Cycle Sequencing Kit (GE Healthcare Life Sciences) following the manufacturer's protocol. Sequences were analyzed using the MegaBACE Sequence Analyzer (v3.0).

## Analysis in human primary chondrocytes

Human primary chondrocytes (mycoplasma-free) were obtained as previously described (Marchini *et al*, 2004). Cells were cultured in DMEM (Gibco) containing 10% FBS (Gibco) and penicillin/strepto-mycin (Gibco) at 37°C, 5% $CO_2$, 95% humidity. To test *CYP26C1* mRNA expression, cells were resuspended in Trizol® (Invitrogen) for RNA preparation according to standard protocols. cDNA synthesis was performed with SuperScript™ II (Invitrogen) according to manufacturer's protocol. Specific primers for *CYP26C1*, *SOX9,* and *SHOX* transcripts were designed (Dataset EV4) and PCR performed. PCR products were confirmed by Sanger sequencing.

To test CYP26C1 protein expression, microsomal preparations were obtained as follows: Cells were washed three times with 2 ml cold solution containing 8 mM $Na_2HPO_4$, 1.5 mM $KH_2PO_4$, and 2.7 mM KCl and subsequently suspended in 1.5 ml of this buffer. Resuspension was centrifuged at 360 *g* for 2 min at 4°C. The pellet was resuspended in 1 ml of 100 mM potassium phosphate buffer pH 7.4 containing 10 mM EDTA. Resuspension was sonicated for $20 \times 1$ s. Lysed cells were then centrifuged at 10,700 *g* for 1 h at 4°C. The pellet was homogenized in 100 μl of 50 mM potassium phosphate buffer pH 7.4, containing 0.1 mM EDTA and 10% glyc-erol (Biomol). Protein concentration was determined using the BCA method. Western blotting experiments were performed following standard protocol with the LI-COR Odyssey® system (LI-COR). hCyp26c1 antibody (PA5-15059, Thermo Fischer Scientific) was diluted as recommended (dilution 1:100) in LI-COR Blocking Buffer (LI-COR), 1× TBS, Tween 0.1%.

*SHOX* expression analysis was performed as follows: Cells were plated in 6-well plates ($1 \times 10^6$ cells per well). After 24 h, cells were treated with ATRA (Sigma-Aldrich, stock solution 1 mM in ethanol absolute) for 6 h and then resuspended in Trizol for RNA extraction according to standard protocols. cDNA synthesis was performed with SuperScript™ II according to manufacturer's protocol. *SHOX* expression was analyzed by quantitative PCR using gene-specific primers and normalized to *HPRT* and *SDHA* (Dataset EV4) with SensiFAST™ SYBR® Lo-ROX Kit (Bioline) in the 7500 Fast Real-Time PCR System (Applied Biosystems).

## Luciferase assays

U2OS cells (human osteosarcoma cells, ATCC; mycoplasma-free) were cultured in DMEM containing 10% FBS and penicillin/strepto-mycin at 37°C, 5% $CO_2$, 95% humidity. Luciferase assays to test *SHOX* variants were performed as follows: Cells were seeded in 24-well plates at $1 \times 10^5$ cells per well. After 24 h, cells were transfected with Lipofectamine 2000 (Invitrogen) according to standard protocols. For each well, cells were transfected with 300 ng of pcDNA4/TO empty vector (Life Technologies), pcDNA4/TO SHOX wild type or mutants; 300 ng of pGL3basic-FGFR3(−3,430/+464) (Decker *et al*, 2011) firefly luciferase reporter construct (Promega); 150 ng pRL-TK *Renilla* luciferase reporter construct (Promega). Mutations were introduced by site-directed mutagenesis (QuickChange Lightning Site-Directed Mutagenesis Kit). After 24 h of transfection, cells were lysed and luciferase activity measured with the Dual Luciferase Assay System (Promega) in a Berthold 96-microplate luminometer. The experiments were performed each time in triplicate.

Luciferase assays to test *CYP26C1* variants were performed as follows: Cells were seeded in 96-well plates at $1 \times 10^4$ cells per well. After 24 h, cells were transfected with Lipofectamine 2000 accord-ing to standard protocols. For each well, cells were transfected with 100 ng of pIRES2-EGFP empty vector (obtained from Dr. Thomas Boettger), pIRES2/EGFP-CYP26C1 wild type or mutants; 100 ng of Cignal RARE Reporter system plasmids (SABiosciences). Mutations were introduced by site-directed mutagenesis. After 24-h transfec-tion, cells were treated with 250 nM ATRA. After 24-h treatment, cells were lysed and assayed with the Dual Luciferase Assay System in a Berthold 96-microplate luminometer. The experiments were performed each time in triplicate.

Luciferase assays to test the *SHOX* promoter were performed as follows: Cells were seeded in 24-well plates at $1 \times 10^5$ cells per well. After 24 h, cells were transfected with Lipofectamine 2000 accord-ing to standard protocols. For each well, cells were transfected with 500 ng of pGL3basic-empty vector (Promega) or pGL3basic-CNE3-SHOX promoter as previously described (Verdin *et al*, 2015); 50 ng of pRL-TK *Renilla* luciferase reporter construct. The CNE3 enhancer was chosen because it was reported with the highest activity on the *SHOX* promoter (Verdin *et al*, 2015). After 24-h transfection, cells were treated with 250 nM ATRA. After 24-h treatment, cells were lysed and assayed with the Dual Luciferase Assay System in a Bert-hold 96-microplate luminometer. The experiments were performed each time in triplicate.

## Zebrafish experiments

Experiments were carried out in wild-type *Danio rerio* strains (AB, TL, Tübingen). Zebrafish were reared according to standard proto-cols. Morpholino were injected at one-cell stage embryos as previ-ously described (Renz *et al*, 2015). The *shox* and *cyp26c1* morpholinos were obtained from Gene Tools (Dataset EV4). Two splicing MOs were designed for *shox* knockdown: MO1 and MO2. In the main text, figures refer to MO1 (Figs 4 and 6). Specific phenotype was confirmed by MO2 (Appendix Fig S3). For *cyp26c1* knockdown, we used a published MO (Liang *et al*, 2012). The standard control MO from Gene Tools was used as a control. Each embryo was injected with 1–2 ng of control, *shox*, or *cyp26c1* MOs. Embryos for double knockdown experiments were injected with 100 pg of *shox* MO and/or 800 pg of *cyp26c1* MO. RNA was extracted from 20–30 injected embryos at 36 h post-fertilization (hpf) with Trizol® accord-ing to standard protocols. cDNA synthesis was performed with SuperScript™ II according to manufacturer's protocol. Effective splic-ing blocking was confirmed with test primers (Dataset EV4) and PCR products were cloned in pSTBlue1 vector (Novagen) and sequenced (Appendix Figs S4–S6). Sense-capped RNA of human *SHOX* and *CYP26C1* wild type and mutants was synthesized using the mMES-SAGE mMACHINE system (Ambion) from pCS2 (Hassel *et al*, 2009).

Expression analyses of *shox* after MO injection were performed as follows: RNA was extracted from 10–15 injected embryos at 55 h post-fertilization (hpf) with Trizol® according to standard protocols.

cDNA synthesis was performed with SuperScript™ II according to manufacturer's protocol. *shox* expression was analyzed by quantitative PCR using gene-specific primers and normalized to *eef1a* and *b-actin* (Dataset EV4) with SensiFAST™ SYBR® Lo-ROX Kit (Bioline) in the 7500 Fast Real-Time PCR System (Applied Biosystems).

Luciferase experiments in zebrafish were performed injecting into one-cell stage embryos doses in the range of 1–2 nl of a 25 nM solution of Cignal RARE Reporter system plasmids and 1–2 ng of control MO or *cyp26c1* MO. After 24-h injection, embryos were separated in groups of 20–30, lysed and assayed with a Dual Luciferase Assay System (Promega) in a Berthold 96-microplate luminometer.

Treatments of zebrafish embryos with RA were performed as follows: Wild-type embryos were left developing for 24 h post-fertilization. At 24 hpf, embryos were separated in groups of 10–15 and treated with mock control or ATRA for 6 h. Finally, RNA was extracted and used for *shox* expression analysis as described above.

## Whole-mount *in situ* hybridization

Whole-mount *in situ* hybridization was performed as described previously (Jowett & Lettice, 1994). Gene-specific primers (Dataset EV4) were used to amplify *col2a1* and *shox* cDNA. Amplicons were then cloned using the psTBlue1 vector (Novagen). Digoxigenin-labeled RNA probes were synthesized using the DIG RNA Labeling Mix (Roche) with the MAXIscript® SP6/T7 Transcription Kit (Ambion). Images were taken with the microscope SZX16, Cell^D Imaging Software (Olympus). Pectoral fin area was measured with Fiji ImageJ (Schindelin *et al*, 2012).

## Statistics

Samples for *in vitro* and zebrafish embryos experiments were randomly assigned to experimental groups, to processing order, and position in multi-well devices. No statistical method was used to predetermine sample size. Group sample sizes for experiments were chosen based on previous studies. Zebrafish embryos that died before analysis were excluded. Statistical analyses were performed using GraphPad Prism version 5 for Windows (GraphPad Software). Data were tested for normality using the D'Agostino and Pearson omnibus normality test. When data fitted normal distribution parametric tests were used, otherwise non-parametric tests were used. Specific tests are described for each group of data in the figure legends. Differences between two groups were analyzed by two-tailed Student's *t*-test. Differences between three or more groups were analyzed by analysis of variance (ANOVA) test. For all experiments, data are expressed as the mean $\pm$ SD. *P*-values < 0.05 were considered significant.

## Bioinformatics resources

Primers for cloning *in situ* hybridization probes, for sequencing, and for testing MO efficacy were designed using Primer3 (Untergasser *et al*, 2012). Primers for quantitative PCR were designed using Universal ProbeLibrary Assay Design Center (Roche). Protein and DNA schemes were drawn using Illustrator of Biological Sequences (IBS) (Liu *et al*, 2015). Mutations were tested with Poly-Phen-2 (Adzhubei *et al*, 2013), Mutation Taster (Schwarz *et al*, 2010), SIFT (Vaser *et al*, 2016), PROVEAN (Choi & Chan, 2015),

### The paper explained

**Problem**
Mutations in the *SHOX* gene cause SHOX deficiency, the most frequent monogenic cause of short stature. The clinical severity of SHOX deficiency varies widely, ranging from short stature without skeletal signs to pronounced skeletal dysplasia. In rare cases, family members with identical mutations even have normal stature. So far, the underlying factors contributing to the phenotypic variability in individuals with SHOX deficiency are unknown. Genetic factors distinct from the *SHOX* gene may represent an important source of clinical variability in SHOX deficiency.
Genetic modifiers are genetic factors that influence the phenotypic expression of a disease-causing gene. Identifying genetic modifiers may result in a better understanding of the clinical variability of SHOX deficiency and enable a more accurate prediction of disease progression.

**Results**
We describe a large family where some individuals with a damaging *SHOX* mutation have normal stature, while other family members with the identical mutation have short stature and skeletal anomalies. To identify the genetic factors that could explain such phenotypic differences, a genetic analysis of this family was performed. This led to the identification of the retinoic acid-degrading enzyme *CYP26C1* as a potential modifying genetic factor. Retinoic acid is the most active biological form of vitamin A and has been shown to play a role in skeleton formation. An excess of this molecule impairs limb development. Expression of *CYP26C1* in human primary chondrocytes (cells that are involved in bone formation) could be demonstrated. *CYP26C1* damaging mutations affect its ability to degrade retinoic acid, leading to higher levels of this vitamin A derivate. High levels of retinoic acid reduced *SHOX* expression in human primary chondrocytes, suggesting that *CYP26C1* regulates *SHOX* expression within the retinoic acid signaling pathway. Two further families with *SHOX* and *CYP26C1* mutations provided further evidence of the most severe phenotype.
Zebrafish represents an animal model particularly suitable to study bone disorders. Decreasing both *SHOX* and *CYP26C1* levels in zebrafish embryos led to smaller fins, further supporting that *SHOX* and *CYP26C1* interact during skeletal development. Mutations in *CYP26C1* result in increased retinoic acid levels which in turn decrease *SHOX* gene expression leading to more severe SHOX deficiency phenotypes.

**Impact**
This study represents an effort to understand the genetic causes of clinical variability. Unraveling factors which modify disease toward milder or more severe phenotypes may lead to novel therapeutic approaches. We provide evidence for *CYP26C1* as a genetic factor influencing SHOX deficiency phenotypic outcomes through the retinoic acid signaling pathway. Manipulating vitamin A metabolism in SHOX deficiency patients may alleviate the skeletal abnormalities of this condition.

and CADD (Kircher *et al*, 2014). Transcription factor binding sites were predicted using PROMO (Messeguer *et al*, 2002; Farré *et al*, 2003).

## Study approval

Studies involving human material were approved by the Review Board Committee at the University Hospital Heidelberg and at the Hamamatsu University School of Medicine. Written informed consent was received from all participants prior to their inclusion in the study. The study was conducted in accordance with the

guidelines of the WMA Declaration of Helsinki and the Department of Health and Human Services Belmont Report. All animal experiments were conducted with approval of local Animal Care Committee and according to institutional guidelines.

**Expanded View** for this article is available online.

## Acknowledgements

We thank Tim Strom, Werner Blum, Miriam Rosilio, Dave Bunyan, Christine Fischer, Slavil Peykov, Herbert Steinbeisser, Thomas Holstein, and Christel Herold-Mende for support. We thank Nagarajan Paramavisam for calculating the CADD scores. Beate Niesler, Simone Berkel, and Antonio Marchini are acknowledged for comments on the manuscript. This study was supported by the DFG (Deutsche Forschungsgemeinschaft), the Medical Faculty of Heidelberg, and the DZHK Heidelberg (German Centre for Cardiovascular Research). AM is a member of the Hartmut Hoffmann-Berling International Graduate School of Molecular and Cellular Biology (HBIGS); GAR is a member of CellNetworks Cluster for Excellence of the University of Heidelberg, Germany. DH and GAR are members of the DZHK.

## Author contributions

GAR designed the project. AM, GAR, and DH designed the experiments and analyzed the data. AM performed the cell culture experiments. LJ and AM performed the experiments in zebrafish embryos. SF-O, GB, MF, TO, and ED recruited the patients and provided DNA for sequencing. AM, RR, and BW performed the sequencing of patients and controls. GN performed whole-genome linkage. AM and GAR wrote the manuscript. AM, RR, DH, and GAR contributed to the discussion and all authors commented the manuscript.

## Conflict of interest

The authors declare that they have no conflict of interest.

## For more information

ExAC
http://exac.broadinstitute.org
TGP
http://browser.1000genomes.org
EVS
http://evs.gs.washington.edu/EVS
PROMO
http://alggen.lsi.upc.es/cgi-bin/promo_v3/promo/promoinit.cgi?dirDB=TF_8.3
SHOX deficiency
http://www.ncbi.nlm.nih.gov/books/NBK1215/
X chromosome gene database
http://grenada.lumc.nl/LOVD2/MR/home.php?select_db=SHOX
OMIM
http://www.omim.org/entry/312865

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
