## [Review Process File · EMBO Molecular Medicine]

Retinoic acid catabolizing enzyme CYP26C1 is a genetic modifier in SHOX deficiency

Antonino Montalbano, Lonny Juergensen, Ralph Roeth, Birgit Weiss, Maki Fukami, Susanne Fricke-Otto, Gerhard Binder, Tsutomu Ogata, Eva Decker, Gudrun Nuernberg, David Hassel, and Gudrun Rappold

Corresponding author: Gudrun Rappold, Institute of Human Genetics

Review timeline:

Submission date:	22 May 2016
Editorial Decision:	13 June 2016
Revision received:	15 September 2016
Editorial Decision:	21 September 2016
Revision received:	28 September 2016
Accepted:	11 October 2016

Transaction Report:

Editor: Céline Carret

1st Editorial Decision

13 June 2016

Thank you for the submission of your manuscript to EMBO Molecular Medicine. We have now heard back from the two referees whom we asked to evaluate your manuscript. Although the referees find the study to be of potential interest, they also raise a number of concerns that need to be addressed in the next final version of your article.

You will see that overall, they both appreciate the in vitro and zebrafish work, but would like to see more definitive evidence regarding the human genetic data (ref.2); I would strongly encourage you to perform the relevant experiments needed to show that the specific variants studied are indeed genetic modifiers in man.

Given these evaluations, I would like to give you the opportunity to revise your manuscript, with the understanding that the referees concerns must be fully addressed and that acceptance of the manuscript would entail a second round of review. Please note that it is EMBO Molecular Medicine policy to allow only a single round of revision and that, as acceptance or rejection of the manuscript will depend on another round of review, your responses should be as complete as possible.

Revised manuscripts should be submitted within three months of a request for revision; they will otherwise be treated as new submissions, except under exceptional circumstances in which a short extension is obtained from the editor.

I look forward to seeing a revised form of your manuscript as soon as possible.

***** Reviewer's comments *****

Referee #1 (Comments on Novelty/Model System):

The authors started with human genetic data on families affected by SHOX deficiency that also carry mutations in CYP26C1 that degrades the important signaling molecule retinoic acid. Then they used human primary chondrocytes to further examine the mechanism to provide evidence that excess retinoic acid is a modifier of SHOX deficiency. Importantly, they also used a zebrafish embryo model to provide very solid support for their novel mechanism of SHOX deficiency. Their results suggest that reducing retinoic acid levels may be beneficial therapeutically for human SHOX deficiency.

Referee #1 (Remarks):

Previous studies on SHOX deficiency in humans have shown that SHOX plays a role in determining height, with some humans carrying SHOX mutations having very short stature, skeletal defects that result in shorter limbs, etc. The authors point out that some humans carrying certain SHOX mutations do not exhibit defects whereas others do, suggesting the existence of modifier genes. Here, they have performed studies using whole genome analysis and whole exome sequencing that identified a CYP26C1 variant that segregates solely in affected SHOX mutant individuals. They carried out genetic and functional assays to show that human CYP26C1 mutations reduce the ability to degrade retinoic acid (RA), and that excess RA reduces SHOX expression. Interestingly, knockdown of either SHOX or CYP26C1 in zebrafish reduces pectoral fin size, with double knockdown being more severe. Thus, they provide solid evidence for a modifying role of CYP26C1 in SHOX phenotype severity that significantly increases our understanding of the diseases caused by SHOX deficiency.

Specific Points to Address:

1. Page 6 in Word doc (top paragraph): After the sentence ending with the Pennimpede et al. 2010 reference, it would be good to add a sentence mentioning that loss of function studies in mouse and zebrafish embryos have shown CYP26C1 is required for RA degradation in hindbrain (Sirbu et al. 2005 Development; Uehara et al. 2007 Dev. Biol.; Hernandez et al. 2007 Development).
2. Page 10 in Word doc (top paragraph): At the end of this paragraph it would be good to clarify the role of RA vs SHOX by saying that the results suggest SHOX expression is not normally affected by endogenous RA, but that loss of CYP26C1 results in excess RA that down-regulates SHOX expression.

Referee #2 (Comments on Novelty/Model System):

Please see my comments to the author about the limitations of the morpholino experiments for proving pathogenicity of a specific genetic variant.

Referee #2 (Remarks):

In this manuscript, Dr. Rappold and colleagues present an intriguing series of experiments linking rare genetic variants in CYP26C1, a retinoic acid catabolizing enzyme, as a potential modifier of the phenotype seen in SHOX deficiency. They first performed linkage analysis followed by exome sequencing in a family with short stature and Madelung deformity and a novel SHOX missense variant that did not segregate with the phenotype in the family. This led to the identification of a novel variant in CYP26C1 which impaired the enzyme's ability to degrade retinoic acid thereby decreasing SHOX activity in in vitro assays. They found 2 other families with rare "damaging" variants in CYP26C1 who also carried mutations in SHOX. Finally, they performed zebrafish morpholino experiments looking at the separate and combined effects of knockdown of these two genes.

The experiments in this manuscript are quite well done and raise the intriguing possibility that variants in CYP26C1 may modify the effects of SHOX mutations. However, I am not certain that

the authors have met the burden of proof to definitively state that this is the case. I have a number of questions which raise concerns that these findings may be coincidental. The crux of the issue is how to define a variant as pathogenic or damaging, either in isolation or in combination with a variant in a second gene.

1. In family 1, could you hypothesize that the F508C variant in CYP26C1 is sufficient by itself to cause the phenotype? The SHOX V161A variant is clearly not sufficient alone as you argue that multiple unaffected family members carry this variant. Is it possible that the SHOX variant is a benign polymorphism? You argue that this variant is damaging based on the evolutionary conservation, in silico prediction models, and the in vitro data. While this is all supportive evidence, there are many examples where all of these items suggest pathogenicity but the variant is not causal. In the in vitro data, the V161A variant appears to be just as deleterious as the known pathogenic positive control variant R153L. If the in vitro assay correctly identifies pathogenicity, then why is the R153L variant capable of producing the SHOX phenotype alone while the V161A variant is not? Of course, this could be due to modifier genes in the various families that you suggest, but it could also be due to the V161A variant not being pathogenic. If you had done a traditional genetics analysis approach where linkage was performed first followed by analysis of the genes in the linkage region, you could have just concluded that the CYP26C1 variant alone could cause the phenotype.

2. Family 2 actually seems to me to potentially be the best evidence of digenicity. The SHOX L132V variant alone does not cause the phenotype. Again, it is possible that the CYP26C1 R378H variant alone could cause the phenotype as we don't know if it is present in the mother who is unaffected. This could be a de novo variant which would then not allow you to support the co-segregation argument. It probably isn't de novo as the same variant was found in the 1000 Genomes database according to your supplementary table. Why don't you know the genotype of this variant in the mother? You clearly had her DNA as you know the SHOX genotype. It would be very important to test the R378H variant in the mother.

3. It is interesting that the L132V variant alone doesn't cause the phenotype. From my review of the reference you cited, it looks like this was previously described in a single family. The prior in vitro data suggests a defect in dimerization and weak DNA binding but normal nuclear localization. This is different from what you state in the manuscript. How convinced are you that the L132V variant is truly pathogenic?

4. The third family really does not provide any definitive information as you state. It is intriguing but the de novo SHOX deletion could easily completely explain the phenotype.

5. So the bottom line about these missense variants for both genes is that they appear to be damaging and require both variants, but there really isn't solid proof of this. To prove it, you will either need a stronger statistical argument (see below) or more animal model work. For the zebrafish work, you have not tested the pathogenicity of any of the individual variants but rather chose a morpholino approach. I understand the rationale for this and it does provide interesting information about the effect of CYP26C1 knockdown but it does not answer the question of the variants' potential pathogenicity. The ideal experiment would be to genome edit each variant into a zebrafish and then edit both a SHOX and CYP26C1 variant into a single fish. If you only saw a subtle or no phenotype with one of the mutations and a distinct phenotype with both, you will have nailed the modifier gene proof. I understand that this is a tremendous amount of work but it would make your argument much stronger. Alternatively, you could at least try to rescue the knockdown with RNA containing the individual gene mutations and demonstrate either a lack of rescue or only partial rescue. You tried to get at this issue with combining the subtherapeutic doses of morpholino. This is nice evidence but given the variability in morpholino effects, it is hard to call this definitive.

6. In terms of statistical proof, family 1 is your discovery cohort. The analysis of family 1 is perfectly appropriate. You then use the additional 68 individuals with SHOX damaging variants and compare them to 140 controls as your validation cohort. You have not done a statistical test to compare the likelihood of finding a damaging CYP26C1 variant in this cohort. If you do a Fisher Exact test on the distribution of

2 0

66 140

Then you get a p-value of ~0.1 (not significant). This could easily just be a power issue. Screening additional individuals with SHOX mutations and gaining statistical proof would greatly strengthen this article. You mention "In addition, no individual from the TGP database was found to carry damaging variants in both genes." How did you do this? Did you use the same definition of damaging (what allele frequency cut off and in silico prediction criteria?). How many people in 1000 Genomes did you look at? This analysis is probably pretty complicated to actually do. You

should be able to make up a statistical likelihood test to look at the rate of co-occurrence of rare potentially damaging mutations in these 2 genes versus CYP26C1 and different genes with a similar length and mutation rate to SHOX. If these 2 genes really interact to cause a phenotype, then there should be a significant decrease in co-occurrence of mutations in these 2 genes. Regardless, you should provide more info about your 1000 Genomes analysis which could significantly strengthen your article.

Regardless of these concerns, the series of experiments are quite interesting and provide a lot of circumstantial evidence that these two genes interact and may augment the phenotype in humans. I think you need to tone down the certainty of your conclusions given all of the caveats noted above. Minor comments:

1. You state, "Heterozygous mutations in its coding or regulatory regions have been identified in up to 10% of patients diagnosed with ISS" This is technically true but I don't think anyone really believes that SHOX mutations account for 10% of ISS. This overstates the problem and also this sentence needs primary references (not the Gene utility card).
2. According to the pattern of transmission of the trait, an autosomal dominant inheritance was hypothesized - You don't mean autosomal, you just mean dominant. SHOX is not autosomal and there is no father to son transmission so you cannot prove it is autosomal.
3. What is "not shown" in the appendix Table S2 for SHOX mutations? This looks weird.
4. In the description of the exome analysis, I would have limited the analysis to the region of linkage and then included a discussion of all the nonsynonymous variants within this region. Those are the ones that by definition will segregate. I wouldn't just limit it to predicted damaging variants as there are many examples of variants which are predicted to be benign actually being the pathogenic variant.

1st Revision - authors' response

15 September 2016

Referee 1

1. Page 6 in Word doc (top paragraph): After the sentence ending with the Pennimpede et al. 2010 reference, it would be good to add a sentence mentioning that loss of function studies in mouse and zebrafish embryos have shown CYP26C1 is required for RA degradation in hindbrain (Sirbu et al. 2005 Development; Uehara et al. 2007 Dev. Biol.; Hernandez et al. 2007 Development).
Agreed and inserted on page 4. Thank you for pointing this out.

2. Page 10 in Word doc (top paragraph): At the end of this paragraph it would be good to clarify the role of RA vs SHOX by saying that the results suggest SHOX expression is not normally affected by endogenous RA, but that loss of CYP26C1 results in excess RA that down-regulates SHOX expression.

We fully agree with this conclusion and now added "Taken together, our results suggest that SHOX expression is not normally affected by endogenous RA in the nanomolar range (de Leenheer et al, 1995; Cunningham et al, 2013); however, loss of CYP26C1 results in an excess RA that down-regulates SHOX expression" (page 9).

Referee 2:

1. In family 1, could you hypothesize that the F508C variant in CYP26C1 is sufficient by itself to cause the phenotype?

This is not likely. The affected individuals show the distinct SHOX-specific skeletal phenotype with a combination of Mesomelia and Madelung deformity. This was the first (clinical) indication that SHOX was involved, subsequently supported by linkage and sequencing studies.

Is it possible that the SHOX V161A variant is a benign polymorphism?

This is very unlikely due to the following reasons:

- a) amino acid resides in highly conserved homeodomain
- b) predicted as damaging by all prediction tools
- c) not present in major databases which argues against a polymorphism
- d) experimentally derived functional data.

Why is the R153L variant capable of producing the SHOX phenotype alone while the V161A variant is not?

This is actually the point. We cannot exclude that R153L in rare cases will produce a non-SHOX deficiency phenotype (if certain modifiers are present), nor can we exclude that the V161A variant alone will lead to a clear-cut SHOX deficiency under specific genetic background.

In this respect it is also interesting to note that rare individuals with normal height and entire *SHOX* gene deletions have been identified (see Introduction page 3 and references Benito-Sanz et al, 2000; Huber et al, 2006; Bunyan et al, 2013), indicating that dependent on the relevant genetic background, SHOX deletions or variants with obvious damaging effects do not always lead to a short stature phenotype.

In family 1, we could demonstrate that the combination of the damaging *SHOX* and *CYP26C1* variants leads to a more severe phenotype. Supportive evidence also comes from family 2, and from our *in vitro* and *in vivo* functional data.

2. Why don't you know the genotype of this variant in the mother? You clearly had her DNA as you know the SHOX genotype. It would be very important to test the R378H variant in the mother.

Unfortunately it is not possible to re-contact the mother of Family 2, and no DNA is left. My lab carried out the SHOX testing on more than 1000 short children including some parents between 2000 and 2004 to identify those with a SHOX mutation/deletion that would be amenable for a clinical trial (Rappold et al, 2007; Blum et al, 2007). Due to the study protocol we have no possibility to re-contact these families.

3. It is interesting that the L132V variant alone doesn't cause the phenotype. From my review of the reference you cited, it looks like this was previously described in a single family. The prior *in vitro* data suggests a defect in dimerization and weak DNA binding but normal nuclear localization. This is different from what you state in the manuscript. How convinced are you that the L132V variant is truly pathogenic?

Thank you very much for this comment! You are entirely correct; we have not correctly cited this. The variant causes a defect in dimerization and DNA binding; nuclear localization was not affected. We have corrected this sentence now on page 7.

We are convinced that the L132V variant is pathogenic as this variant was also previously identified in two affected individuals of a Swedish family with LWD (Grigelioniene et al, Hum Genet 107:145-149, 2000). The variant resides in an absolutely conserved region of the homeodomain, was predicted as damaging and was functionally verified.

4. The third family really does not provide any definitive information as you state. It is intriguing but the *de novo* SHOX deletion could easily completely explain the phenotype.

Agreed; but it adds to the overall evidence that damaging variants in SHOX and CYP26C1 can co-occur in individuals with severe LWD phenotypes (page 6).

5. So the bottom line about these missense variants for both genes is that they appear to be damaging and require both variants, but there really isn't solid proof of this. To prove it, you will either need a stronger statistical argument (see below) or more animal model work.

A) Animal work:

The experiments using different subphenotypic dosages of *shox* and *cyp26c1* MOs have already given strong evidence. While single knockdown of either *shox* or *cyp26c1* using subphenotypic MO doses did not result in any obvious phenotype, double knockdown (simultaneous injection) of *shox* and *cyp26c1* produced significantly smaller pectoral fins (Fig. 6A-D). The relative expression of *shox* mRNA expression was also significantly reduced and staining for *col2a1* revealed impaired pectoral fin development.

But in Appendix Figures S7 and S8 we now provide additional functional data of the *SHOX* and *CYP26C1* variants in zebrafish embryos. We performed morpholino knockdown experiments combined with RNA injection, but the embryos, due to high toxicity displayed strong developmental aberrations. Injection of the RNA alone, however, leads to interesting phenotypes:

Injection of capped-RNA encoding human wild type *SHOX* resulted in longer pectoral fins compared to controls and compared to the *SHOX* V161A (family 1) and L132V (family 2) variants, further corroborating the pathogenicity of these variants (Appendix Fig S7). (Note that the patient in family 3 bore a *SHOX* deletion.) Embryos overexpressing *SHOX* wild type consistently presented a stronger *col2a1* signal as shown by *in situ* hybridization analysis (Appendix Fig S7B and C). These

results are in agreement with a previous report which demonstrated that overexpression of *Shox* in chicken leads to longer skeletal elements and a delayed timing of ossification (Tiecke et al, 2006). Injection of capped-RNA encoding human wild type *CYP26C1* resulted in the absence of the pectoral fins (Appendix Fig S8). We hypothesize that ectopic expression of *CYP26C1* wild type leads to retinoic acid deficiency. Deficiency of retinoic acid in the early stages of pectoral fin development has been previously shown to lead to similar phenotypes (Begemann et al, 2001). Overexpression of the variants identified in families 1-3 did not significantly alter pectoral fins development. Therefore, together with our luciferase assays in U2OS cells (Fig 2), these results strongly suggest that the identified variants affect the ability of *CYP26C1* to degrade retinoic acid, further corroborating their potential pathogenicity.

B) Stronger statistical proof: please see point 6

6. In terms of statistical proof, family 1 is your discovery cohort. The analysis of family 1 is perfectly appropriate. You then use the additional 68 individuals with *SHOX* damaging variants and compare them to 140 controls as your validation cohort. You have not done a statistical test to compare the likelihood of finding a damaging *CYP26C1* variant in this cohort. If you do a Fisher Exact test on the distribution of 2 0 66

140 Then you get a p-value of ~0.1 (not significant). This could easily just be a power issue. Screening additional individuals with *SHOX* mutations and gaining statistical proof would greatly strengthen this article.

Thank you for this comment. As we had no access to further patients with damaging *SHOX* mutations, we screened further 210 healthy controls with normal height, and no variant affecting *CYP26C1* activity was identified (Appendix Table S3; Appendix Fig S2). We also state on page 5 “No functionally damaging variants in *CYP26C1* were found in the control individuals with normal height (p -value = 0.0261, two-tailed Fisher’s exact test).”

You mention "In addition, no individual from the TGP database was found to carry damaging variants in both genes." How did you do this? Did you use the same definition of damaging (what allele frequency cut off and in silico prediction criteria?). How many people in 1000 Genomes did you look at? This analysis is probably pretty complicated to actually do. You should be able to make up a statistical likelihood test to look at the rate of co-occurrence of rare potentially damaging mutations in these 2 genes versus *CYP26C1* and different genes with a similar length and mutation rate to *SHOX*. If these 2 genes really interact to cause a phenotype, then there should be a significant decrease in co-occurrence of mutations in these 2 genes. Regardless, you should provide more info about your 1000 Genomes analysis which could significantly strengthen your article.

To estimate the frequencies we summed up the allele frequencies of each damaging variant (predicted as damaging by at least one of the prediction tools used) reported in the ExAC database. We now state in the Discussion, page 8 “In addition, we browsed TGP for genotypes bearing both *SHOX* and *CYP26C1* damaging variants (predicted as damaging by at least one out the prediction tools used). A limitation of this approach is that not for all variants individual genotypes are available. We could not find any individual in the TGP database carrying damaging variants in both genes (1000 Genome Project Consortium, 2015). Finally, we browsed the ExAC database and estimated the frequency of *SHOX* and *CYP26C1* damaging variants (Exome Aggregation Consortium, 2015). The estimated allele frequencies in the ExAC database were 0.3% and 0.6% for *SHOX* and *CYP26C1*, respectively. With these estimated frequencies, the probability for an individual to bear a damaging variant in both genes is 1.8×10^{-5} .”

Regardless of these concerns, the series of experiments are quite interesting and provide a lot of circumstantial evidence that these two genes interact and may augment the phenotype in humans. I think you need to tone down the certainty of your conclusions given all of the caveats noted above.

We have toned down the conclusions in the Discussion on page XX as proposed by this Referee “as biological triggers that we **suggest** to modify the severity of *SHOX* deficiency...”.

Minor comments:

1. You state, "Heterozygous mutations in its coding or regulatory regions have been identified in up to 10% of patients diagnosed with ISS" This is technically true but I don't think anyone really believes that *SHOX* mutations account for 10% of ISS. This overstates the problem and also this sentence needs primary references (not the Gene

utility card).

We agree that the Gene utility card is not the best reference and now included a recent review reference by Marchini et al., 2016. Table 2 in this review summarizes all 15 publications on the prevalence in patients with isolated/idiopathic short stature which indicates that the combination of MLPA and sequencing leads to a frequency between 8 to 22%. Thus we think that “up to 10% of patients diagnosed with ISS” is not overstated.

2. According to the pattern of transmission of the trait, an autosomal dominant inheritance was hypothesized - You don't mean autosomal, you just mean dominant. SHOX is not autosomal and there is no father to son transmission so you cannot prove it is autosomal.

Agreed and changed accordingly in the manuscript.

3. What is "not shown" in the appendix Table S2 for SHOX mutations? This looks weird.

Details on specific *SHOX* variants were given only for the variants identified in family 1, 2, and 3. Other individual *SHOX* variants are not shown.

Rational: Appendix Table S2 lists all the identified *CYP26C1* variants including common variants and polymorphisms in our cohort of 68 LWD individuals. Some of the common variants were present more than once in this cohort. Introducing all the details for each *SHOX* variant associated with the respective *CYP26C1* variant would have resulted in a very large table. For example, variant 4 (p.Thr213=) was found 40 times in our cohort. In column “*SHOX* variant” we would have had to indicate 40 different entries, associate row number 4 to the *CYP26C1* variant p.Thr213= which would be quite difficult to read.

4. In the description of the exome analysis, I would have limited the analysis to the region of linkage and then included a discussion of all the nonsynonymous variants within this region. Those are the ones that by definition will segregate. I wouldn't just limit it to predicted damaging variants as there are many examples of variants which are predicted to be benign actually being the pathogenic variant.

Yes, this would have been a possibility, but we decided not to limit ourselves to the results from the linkage analysis in order to have an unbiased approach.

2nd Editorial Decision

21 September 2016

Thank you for the submission of your revised manuscript to EMBO Molecular Medicine. We have now received the enclosed report from the referee who was asked to re-assess it. As you will see, the reviewer is now supportive and I am pleased to inform you that we will be able to accept your manuscript pending final editorial amendments.

Please submit your revised manuscript within two weeks.

***** Reviewer's comments *****

Referee #2 (Remarks):

The authors have done an excellent job addressing my prior concerns. The overexpression of the individual variants in the zebrafish is very nice additional evidence. While the human genetics evidence is not definitive, there is substantial amount of supportive evidence to suggest that the authors are correct in their assertion that damaging variants in *CYP26C1* modify the *SHOX* genotype. As these variants are only present in a minority of carriers of *SHOX* mutations, there must be additional modifier genes. It will be interesting to see if additional families can be found with clear evidence of modifying variants in the retinoic acid metabolism pathway.

Corresponding Author Name: Gudrun A. Rappold

Manuscript Number: EMM-2016-06623-V2